# Controlled Release of the α-Tocopherol-Derived Metabolite α-13′-Carboxychromanol from Bacterial Nanocellulose Wound Cover Improves Wound Healing

**DOI:** 10.3390/nano11081939

**Published:** 2021-07-28

**Authors:** Jessica Hoff, Berit Karl, Jana Gerstmeier, Uwe Beekmann, Lisa Schmölz, Friedemann Börner, Dana Kralisch, Michael Bauer, Oliver Werz, Dagmar Fischer, Stefan Lorkowski, Adrian T. Press

**Affiliations:** 1Department of Anesthesiology and Intensive Care Medicine, Jena University Hospital, Am Klinikum 1, 07747 Jena, Germany; jessica.hoff@med.uni-jena.de (J.H.); michael.bauer@med.uni-jena.de (M.B.); 2Center for Sepsis Control and Care (CSCC), Jena University Hospital, Am Klinikum 1, 07747 Jena, Germany; 3Pharmaceutical Technology and Biopharmacy, Institute of Pharmacy, Friedrich Schiller University Jena, Lessingstraße 8, 07743 Jena, Germany; berit.karl@uni-jena.de (B.K.); beekmann@jenacell.de (U.B.); dana.kralisch@uni-jena.de (D.K.); dagmar.fischer@fau.de (D.F.); 4Department of Pharmaceutical and Medicinal Chemistry, Institute of Pharmacy, Friedrich Schiller University Jena, Philosophenweg 14, 07743 Jena, Germany; jana.gerstmeier@uni-jena.de (J.G.); friedemann.boerner@uni-jena.de (F.B.); oliver.werz@uni-jena.de (O.W.); 5Institut of Nutritional Sciences, Friedrich Schiller University Jena, Dornburger Straße 25, 07743 Jena, Germany; lisa.schmoelz@uni-jena.de; 6Competence Cluster for Nutrition and Cardiovascular Health (nutriCARD) Halle-Jena-Leipzig, Dornburger Straße 25, 07743 Jena, Germany; 7Jena Center for Soft Matter (JCSM), Friedrich Schiller University Jena, Philosophenweg 7, 07743 Jena, Germany; 8Department of Chemistry and Pharmacy, Divison of Pharmaceutical Technology, Friedrich Alexander University Erlangen-Nürnberg, Cauerstrasse 4, 91058 Erlangen, Germany; 9Medical Faculty, Friedrich Schiller University Jena, Bachstr. 18, 07743 Jena, Germany

**Keywords:** biocompatible nanocellulose, wound healing, drug delivery, controlled release, inflammation, tocopherols, natural products

## Abstract

Inflammation is a hallmark of tissue remodeling during wound healing. The inflammatory response to wounds is tightly controlled and well-coordinated; dysregulation compromises wound healing and causes persistent inflammation. Topical application of natural anti-inflammatory products may improve wound healing, in particular under chronic pathological conditions. The long-chain metabolites of vitamin E (LCM) are bioactive molecules that mediate cellular effects via oxidative stress signaling as well as anti-inflammatory pathways. However, the effect of LCM on wound healing has not been investigated. We administered the α-tocopherol-derived LCMs α-13′-hydroxychromanol (α-13′-OH) and α-13′-carboxychromanol (α-13′-COOH) as well as the natural product garcinoic acid, a δ-tocotrienol derivative, in different pharmaceutical formulations directly to wounds using a splinted wound mouse model to investigate their effects on the wounds’ proinflammatory microenvironment and wound healing. Garcinoic acid and, in particular, α-13′-COOH accelerated wound healing and quality of the newly formed tissue. We next loaded bacterial nanocellulose (BNC), a valuable nanomaterial used as a wound dressing with high potential for drug delivery, with α-13′-COOH. The controlled release of α-13′-COOH using BNC promoted wound healing and wound closure, mainly when a diabetic condition was induced before the injury. This study highlights the potential of α-13′-COOH combined with BNC as a potential active wound dressing for the advanced therapy of skin injuries.

## 1. Introduction

Inflammation is a hallmark of the immune defense and tissue remodeling to promote wound healing after injury [1]. The inflammatory response to wounds is a tightly controlled and well-coordinated process, where different types of immune cells promote wound healing. In particular, fibroblasts stimulate the early proliferative phase during the healing process, while macrophages support the regeneration through growth factors [2,3]. Eicosanoids are bioactive lipid mediators that recruit immune cells, thereby promoting inflammation [4]. On the other hand, eicosanoids modulate resident cells and keratinocyte activity, contributing to wound closure and scar formation [5]. Dysregulation of these processes compromises wound healing, for instance, during diabetes, and causes persistent inflammation leading to increased wounds with complications and defective healing.

The skin is the first barrier of the body for microbial invaders, and lesions need to be closed and repaired quickly to reduce the risk of infection. In patients with diabetes, however, the healing process is disturbed and slowed down. Especially the re-epithelialization of the skin is significantly delayed in diabetic patients [6]. It is speculated that the exposure to high glucose concentrations renders keratinocytes and other immune cells insensitive to growth factors, thus leading to a dysregulated healing process [7,8,9]. However, anti-inflammatory substances, including plant-derived compounds such as tocopherols, may be applied to treat inflammation.

Tocopherols are an effective form of vitamin E, a term that usually describes a group of lipid-soluble molecules with anti-oxidative and non-anti-oxidative properties. Tocopherols are known to be antioxidants with a proven independent anti-inflammatory effect [10,11]. More recently, it has been shown that vitamin E deficiency may contribute to the onset of degenerative disorders, immune pathologies, and even atherosclerosis [12,13,14]. Often, vitamin E is a component in food additives, but their precise mechanistic action in disease promotion, modulation, and suppression has not been entirely conceptualized. In recent placebo-controlled randomized trials, vitamin E supplementation positively affected the treatment of non-alcoholic fatty liver disease and Alzheimer’s disease [15,16,17,18].

In the body, vitamin E is rapidly metabolized in the liver through ω- and β-oxidation leading to long-chain metabolites that differ in the oxygen content of functional chemical groups (for α-tocopherol: α-13′-hydroxychromanol (α-13′-OH) or α-13′-carboxychromanol (α-13′-COOH)) [19]. These metabolites undergo shortening of their side chain to form the water-soluble end-product α-carboxyethylhydroxychromanol. Just recently, it has been shown that α-13′-COOH inhibits 5-lipoxygenase and microsomal prostaglandin E_2_ synthase (mPGES-1), both critical enzymes in proinflammatory eicosanoid biosynthesis [20,21]. Derivatives of α-tocopherols, namely, α-13′-OH and α-13′-COOH, have been detected in plasma with concentrations of 0.8 and 1.2 nmol L^−1^ under physiological concentrations, but plasma concentrations are increased in humans and rodents upon supplementation of supraphysiological doses [22]. The physiological role of the tocopherol metabolites and their modes of action are still unknown and need to be elucidated. However, the physiological occurrence of α-13′-OH and α-13′-COOH and their anti-inflammatory properties grant a potential for their pharmaceutical applications.

In therapeutic concentrations, tocopherols can reduce inflammation signs and support the healing process in wounds. These substances can be directly loaded into the dressing used for covering the wound. The biopolymer bacterial nanocellulose (BNC) is known for its suitability as a drug carrier and a wound dressing [23]. BNC is biosynthesized by bacteria, e.g., the bacterial strain *Komagataeibacter xylinus*, and offers a unique nano-sized network with high surface area allowing high drug load and efficient drug delivery. Due to its outstanding physicochemical and biological characteristics, it is already widely used for medical applications, such as implants, scaffolds, or wound dressings [24]. Nowadays, BNC has become a clinical standard wound cover that improves wound healing, particularly for complicated wounds [25]. Purity, biocompatibility, non-allergenicity, capability to provide moisture and absorb wound exudate, gas permeability, and painless removal are just a few of the many favorable properties of BNC, facilitating its utilization as a wound dressing [26]. In addition, BNC is loaded with pharmaceutical ingredients to create active wound dressings, which allow the delivery of drugs such as antibiotics that may further support the treatment of infected wounds [27]. Controlled delivery of tocopherol metabolites, known as natural modulators of 5-LO and mPGES-1 activity, from the BNC dressing could actively support the wound healing process and shorten the healing time.

This study investigated the effects of tocopherol metabolites and garcinoic acid, a natural product obtained from *Garcinia kola* seeds and δ-tocotrienol-derived long-chain metabolites, in different pharmaceutical formulations directly administered to the wound and loaded into BNC on wound healing under normal and diabetic conditions.

## 2. Materials and Methods

### 2.1. Isolation, Semi-Synthesis, and Dilution of Metabolites

*Garcinia kola* seeds were a gift from AnalytiCon Discovery (Potsdam, Germany). Isolation of garcinoic acid (GA) from *Garcinia kola* seeds and its purification were performed as reported recently [28].

α-13′-COOH was synthesized from GA as described [29]. The purity of GA and α-13′-COOH used was higher than 95%, as confirmed using an LC-MS/MS system comprised of a Dionex UltiMate 3000 UHPLC system (Thermo Fisher Scientific, Waltham, MA, USA) coupled to a Bruker amaZon SL Ion trap mass spectrometer (Bruker, Billerica, MA, USA) equipped with an atmospheric pressure chemical ionization (APCI) source (Bruker, Billerica, MA, USA) as described.

Natural metabolites (α-13′-OH, α-13′-COOH, GA) were dissolved in poly(ethylene glycol) 400 (PEG400, Carl Roth, Karlsruhe, Germany) at 1, 5, or 10 µmol L^−1^ under gentle agitation on an orbital shaker (IKA KS 400 ic control, Staufen im Breisgau, Germany) at room temperature. A loading solution was prepared for loading the BNC fleeces by dissolving α-13′-COOH directly in PEG400 at 0.16 µg mL^−1^. The vehicle solution was composed of 100% PEG400.

### 2.2. Animals

All experimental procedures with animals have been approved by the ethical committee and local government authority of Thuringia, Germany (Thüringer Landesverwaltungsamt, Registration Number: UKJ-17-035, 11 October 2017). Eight- to ten-week-old FVB/N mice (male and female) were used for all experiments (n = 4 to 6 per group). All animals were housed in the Central Experimental Animal Facility of the Jena University Hospital, Jena, Germany, providing a specific pathogen-free environment, constant humidity (50 to 60%), a 12 h light/dark cycle, and a constant ambient temperature (24 °C). Access to rodent chow and drinking water ad libitum was always ensured.

### 2.3. Diabetes Mouse Model

Mice received 135 mg per kg body weight streptozotocin (STZ, Biomol, Hamburg, Germany) in freshly prepared citrate buffer (0.1 mol L^−1^ sodium citrate dehydrate (Carl Roth, Karlsruhe, Germany), 0.1 mol L^−1^ citric acid (Carl Roth, Karlsruhe, Germany), pH 4.5) by intraperitoneal injection once to induce diabetes (Appendix A, Appendix A). Blood glucose concentrations and body weight were measured every two days. Surgical intervention to generate skin wounds was performed two weeks after diabetes induction.

### 2.4. Splinted Wound Model

Twenty-four hours before surgery, analgesics were added to the drinking water, supplying a daily dose of approximately 100 mg metamizole (Novaminsulfon, Ratiopharm, Ulm, Germany) per kg body weight per day to the average water intake of the mice. The analgesic regime was maintained post-operative till the end of the experiment.

All surgical interventions were carried out under general isoflurane (CP-Pharma, Burgdorf, Germany) anesthesia in a semi-sterile environment with tip-sterilized surgical instruments and sterilized materials. After removing the hair from the back of the mice, two 5 mm full-thickness excisional circular skin-wounds were generated using a biopsy punch (Stiefel, München, Germany). Animals treated with PEG400-diluted tocopherol metabolites or GA received self-made, sterilized silicon rings (inner diameter 5 mm, outer diameter 10 mm) directly onto the wound and were fixed by 5-0 braided silk sutures (Ethicon, Somerville, NJ, USA). The compounds were administered in a volume of 40 µL on the wound. An occlusive dressing (Raucodrape Surgical Incise Drape, Lohmann & Rauscher, Rengsdorf, Germany) was placed tightly on the top of the splint ring to prevent the solution from leaking. The wound dressing and splinting ring were omitted for groups treated with bacterial nanocellulose pads (BNC groups). Instead, the BNC pads were fixed directly with four interrupted sutures on top of the wounds and served as splinting rings. All animals were housed in individual cages after the surgery and monitored daily for signs of inflammation and pain. The solutions and wound dressing or BNC were replaced daily. The animals were sacrificed by heart puncture after 10 days under general anesthesia. The wounds were excised with the wound margins from each mouse. The tissue was fixed in 4% formalin (ROTI-Histofix, Carl Roth, Karlsruhe, Germany) and embedded in paraffin for immunohistochemistry.

### 2.5. Histopathological Analysis

Five-micrometer sections were deparaffinized in xylene (Carl Roth, Karlsruhe, Germany), rehydrated in a series of ethanol (Carl Roth, Karlsruhe, Germany), and stained with hematoxylin (Carl Roth, Karlsruhe, Germany) and eosin (Merck, Berlin, Germany) (H&E) to detect dermal tissue and epidermal recovery. Therefore, the thickness of the epidermis was measured with five independent lanes. For polymorphonuclear neutrophils (PMN) staining, skin sections were deparaffinized and afterward permeabilized in citrate buffer pH 6 for 25 min (Dako, Santa Clara, CA, USA). Staining was carried out by incubating tissue sections in PBS (Lonza, Basel, Switzerland) containing 5% donkey serum (Equitec-Bio, Kerrville, TX, USA) (37 °C, 30 min), following an overnight incubation at 4 °C with a primary antibody against PMN (Gentaur, Kampenhout, Belgium) diluted 1:500 in antibody diluent (Dako, Santa Clara, CA, USA). Negative control was incubated with antibody diluent. After further washing steps, the slides were then incubated with Alexa-Fluor 568 fluorescently labeled anti-rabbit IgG antibody (1:500, Thermo Fisher Scientific, Waltham, MA, USA) for 20 min at room temperature. Nuclei were counterstained with Hoechst 33342 (1:2000, Sigma-Aldrich, St. Louis, MO, USA). Finally, the sections were mounted with a ROTI-Mount FluorCare mounting media (Carl Roth, Karlsruhe, Germany). Images of areas of each slide with a magnification of 20× were acquired using an LSM 780 microscope (Carl Zeiss, Jena, Germany). Quantification of stained cells in the sections was performed by counting the total number of positive cells in the picture using ImageJ version 1.51 (N.I.H. Freeware, Bethesda, MD, USA).

### 2.6. Collagen Quantification by Second Harmonic Generation Imaging

Second-harmonic generation of two-photon fluorescence imaging was performed using an LSM 780 microscope (Carl Zeiss, Jena, Germany). Two-photon excitation at 800 nm (Chameleon Nd:YAG Laser, Coherent) was used to generate a collagen-specific second harmonic frequency at 430 nm, which was detected on a non-descanned detector through a 375 nm to 425 nm band-pass. Z-stacks were recorded and analyzed with a 20× plan-apochromatic objective (NA: 0.8).

### 2.7. Lipid Mediator Analysis Using UPLC-MS/MS

The dissected wound or healthy skin was homogenized in methanol (Thermo Fisher Scientific, Waltham, MA, USA) (50 mg tissue in 200 µL) and kept overnight at −20 °C. Precipitated proteins were removed by centrifugation (13,000 g, 5 min, 4 °C), and the supernatant was diluted with 2 mL of methanol (Thermo Fisher Scientific, Waltham, MA, USA) and water (50:50, *v*/*v*) containing the deuterium-labeled internal standards d4-PGE_2_ and d5-RvD2 (500 pg each, Cayman Chemical, Hamburg, Germany). Samples were kept at −20 °C for 60 min and then centrifuged (1200 g, 4 °C, 10 min). Solid-phase C18 cartridges (Sep-Pak Vac 6cc, Waters, Milford, MA, USA) were equilibrated with 6 mL methanol (Thermo Fisher Scientific, Waltham, MA, USA) before adding 2 mL water. Next, 9 mL acidified water (pH 3.5, HCl) was added to the samples and then loaded onto the conditioned C18 columns. The columns were washed once with 6 mL water, followed by 6 mL *n*-hexane (Thermo Fisher Scientific, Waltham, MA, USA). The lipid mediators were eluted with 6 mL of methyl formate (Thermo Fisher Scientific, Waltham, MA, USA). Samples were brought to dryness using an evaporation system (TurboVap LV, Biotage, Uppsala, Sweden) and immediately suspended in methanol/water (50:50, *v*/*v*) UPLC-MS/MS measurement. The UPLC-MS/MS system consisted of an ACQUITY UPLC BEH C18 column (1.7 µm, 2.1 mm × 50 mm, Waters, Milford, MA, USA) and an ACQUITY TM UPLC (Waters, Milford, MA, USA) as well as a QTRAP 5500 Mass Spectrometer equipped with an electrospray ionization source (Sciex, Darmstadt, Germany). The QTRAP 5500 was operated in negative ionization mode using scheduled multiple reaction monitoring (MRM) coupled with the information-dependent acquisition (IDA) and enhanced production scan (EPI). The scheduled MRM window was 90 s. Each lipid mediator parameter was optimized individually. Using an MRM method with diagnostic ion fragments and identification, lipid mediator analysis was performed as described previously [30,31].

### 2.8. Cell Culture

Primary human fibroblasts were kindly provided by the Institute of Human Genetics, University Hospital Jena. The cells were cultured in high glucose (4.5 g L^−1^) Dulbecco’s modified Eagle’s medium (DMEM, Sigma-Aldrich, Saint Louis, MO, USA) containing 10% (*v*/*v*) fetal bovine serum (FBS, Biochrom/Merck, Berlin, Germany) and 1% (*v*/*v*) L-glutamine-penicillin-streptomycin solution (Sigma-Aldrich, Saint Louis, MO, USA). For sub-culturing, cells were detached by trypsin-EDTA (Sigma-Aldrich, Saint Louis, MO, USA) treatment and were split weekly (1:5 split). HaCaT keratinocytes were cultured in high glucose (4.5 g L^−1^) DMEM containing 10% (*v*/*v*) FBS and 1% (*v*/*v*) L-glutamine-penicillin-streptomycin solution. For sub-culturing, cells were detached by trypsin treatment and were cultivated twice per week (1:5 to 1:10 split).

### 2.9. Wound Healing Environment

For the analysis of gene expression involved in wound healing and proliferation, the in vivo conditions of wound healing were imitated in vitro using fibroblasts and keratinocytes treated with 10 ng mL^−1^ TNF-α (Sigma-Aldrich, Saint Louis, MO, USA)and 5 ng mL^−1^ IL-1β (Sigma-Aldrich, Saint Louis, MO, USA) as in vitro wound healing conditions [32].

### 2.10. RNA Isolation and cDNA Synthesis

Total RNA from fibroblasts and keratinocytes was isolated using the Qiagen RNeasy Mini kit (Qiagen, Hilden, Germany). According to the manufacturer’s protocol, cDNA synthesis was performed using the RevertAid First Strand cDNA synthesis kit (Thermo Fisher Scientific, Waltham, MA, USA) and 500 mg L^−1^ oligo-dT primers.

### 2.11. Quantitative Real-Time RT-qPCR

RT-qPCR was run on a LightCycler 480 instrument (Roche Diagnostics, Mannheim, Germany) using Maxima SYBR Green qPCR Master Mix (Thermo Fisher Scientific, Waltham, MA, USA) as described [33]. Primers (COL1A1, MMP2, TIMP1, TIMP2, TIMP3) were purchased from Invitrogen (Karlsruhe, Germany) (Appendix A). PCR results were analyzed using the LightCycler software version 1.5.0.39 (Roche Diagnostics, Mannheim, Germany).

### 2.12. Scratch Assay

Cells were seeded on 8-well slides (5300 fibroblasts per well or 18,500 keratinocytes per well, BD Falcon, Franklin Lakes, NJ, USA). Cells were then treated with the indicated substances and stained with hematoxylin (Sigma-Aldrich, Saint Louis, MO, USA) after 4, 24, and 48 h. After 48 h, cells were scratched with a 1 mL pipette tip and carefully washed.

### 2.13. Preparation of Bacterial Nanocellulose

Bacterial nanocellulose (BNC) was produced by cultivation of *Komagataeibacter xylinus* (*K. xylinus*) strain DSM 14666, deposited at the German Collection of Microorganism and Cell Cultures (DSMZ, Braunschweig, Germany). For cultivation, a preculture of *K. xylinus* in the Hestrin–Schramm culture medium (HSM) was used [34]. HSM contains 2% glucose (Carl Roth, Karlsruhe, Germany), 0.5% peptone (Carl Roth, Karlsruhe, Germany), 0.5% yeast extract (Carl Roth, Karlsruhe, Germany), 0.34% disodium hydrogen phosphate (VWR, Radnor, PA, USA) and 0.115% citric acid (Carl Roth, Karlsruhe, Germany). One liter of preculture was added to 4 L HSM and maintained for 7 days at 28 °C in a pilot-scale plant (JeNaCell, Jena, Germany). Static cultivation was used, characterized by forming a homogenous three-dimensional nanostructured cellulose network at the boundary layer between medium and air over time [23,35]. The BNC fleece thus obtained was subsequently harvested and deposited in 0.1 M sodium hydroxide solution (Carl Roth, Karlsruhe, Germany) overnight and washed several times afterward with water for injection until pH neutrality. Finally, the purified BNC was mechanically cut into 1.9 cm^2^ round pellicles and sterilized by autoclaving (121 °C, 20 min, 2 bar) for further experiments.

### 2.14. Dimensions of the Bacterial Nanocellulose

Cylindrically shaped BNC fleeces were characterized regarding their dimensions and weight. Weight was determined with a mass balance (Sartorius, Goettingen, Germany). A caliper (Wiha Werkzeuge, Schonach im Schwarzwald, Germany) was utilized to investigate the height and diameter of samples. Volume and surface area were calculated with the geometrical equations of a circular cylinder as described earlier [36].

### 2.15. Loading of Bacterial Nanocellulose

For the preparation of the loading solution, α-13′-COOH was dissolved in PEG400 (Carl Roth, Karlsruhe, Germany) at 0.16 µg mL^−1^. The BNC fleeces were immersed in a 10 mL loading solution in airtight closed containers at room temperature for 48 h. The process was conducted under light exclusion and shaking on a temperature-controlled orbital shaker (IKA KS 400 ic control, Staufen im Breisgau, Germany) at 70 rpm. Drug-free, PEG400-loaded BNC fleeces were loaded with 100% PEG400 (BNC-PEG400).

### 2.16. Moisture Retention Capacity

First, the weight of BNC samples was determined with a mass balance (Sartorius, Goettingen, Germany) (0 h). The BNCs were placed afterward on a non-absorbent matrix without coverage at room temperature under a light exclusion for 144 h. Weight measures after 1, 2, 4, 6, 8, 24, 72, and 144 h were taken to calculate relative weight to the weight at time point 0 h. 

### 2.17. Statistics

Data are presented as mean ± standard deviation (S.D.) or mean ± standard error (S.E.) of independent experiments or animals as indicated in the figure legends. Data analysis was performed using GraphPad Prism version 8 (San Diego, CA, USA) and SigmaPlot version 14.0 (Systat Software, San Jose, CA, USA). The figure legends describe the number of replicates and the statistical tests where indicated. 

## 3. Results

In the splinted wound model, a standard model to study wound healing in early experimental phases in mice, a circular skin wound of 0.5 cm in diameter was created and kept open by a splinting silicon ring [37]. These wounds predominantly heal through cell migration and proliferation rather than from contraction [37,38]. The time for complete wound closure in mice requires about 10 days (Figure 1A). In this model, we tested different tocopherol and tocotrienol metabolites with known anti-inflammatory properties (α-13′-hydroxychromanol, α-13′-OH; α-13′-carboxychromanol, α-13′-COOH) and GA on the temporal wound healing process. We formulated the different metabolites first in PEG400 and applied them on the wound once per day at two doses (1, 5, or 10 µmol L^−1^). While α-13′-OH showed no effects, GA and especially α-13′-COOH significantly accelerated wound healing in the late phase (after day 5) in a dose-dependent manner (Figure 1C–E), whereas the vehicle solution PEG400 showed no effects on wound healing when compared to wounds without treatment (Figure 1B).

After 10 days of observation, animals were sacrificed, and wounds were prepared for histological analysis (Figure 2A). The lower concentrations used for all substances did not affect the structure of the wound tissue and newly generated skin (Appendix A). At higher doses, histological analysis revealed a significant thickening of the epidermis in α-13′-OH-treated animals, which was also observed, but less pronounced in α-13′-COOH-treated animals. GA showed no effect on epidermal thickness (Figure 2B). Further, the number of neutrophils that remained in the wound bed was increased upon α-13′-OH treatment, suggesting increased recruitment and prolonged proinflammatory phase in these animals. Neutrophil counts in the other treatment group remained unaltered or slightly reduced (Figure 2C). Increased collagen deposition is a sign of scar formation, which is an unfavorable condition as, especially in more extensive wounds, scars destabilize the skin, which is a sign of defective healing. Two-photon microscopy excitation of collagen’s second harmonic generation detects polymerized deposited collagen fibers. The newly built collagen network underneath the wound-closing first layer was quantified from two-photon microscopic images (Figure 2D). All treatment groups showed tendencies of decreased amounts of collagen in their wound. However, the changes were not statistically significant, suggesting no excessive collagen deposition and defective healing due to the treatment (Figure 2D). Further, α-13′-COOH reduced the wound-induced activation of proinflammatory cyclooxygenase (COX), as reflected by reducing COX-derived prostaglandin (PG)E_2_ and PGD_2_, a finding that is associated with improved wound healing (Figure 2E) [39]. Interestingly, increased levels of the specialized pro-resolving mediator resolvin (Rv) D5, maresin 1 (MaR1), and protectin DX (PDX) accompanied the reduction of proinflammatory lipid mediators in these animals (Figure 2E).

The findings in the animal model correlate with the response of primary fibroblasts and keratinocytes (HaCaT cells) to α-13′-COOH in a proinflammatory wound healing environment (WHE; 5 µg L^−1^ IL-1β, 10 µg L^−1^ TNF-α). In fibroblasts, α-13′-COOH increased the expression of metalloproteinase matrix metalloproteinase-2 (MMP2) and tissue inhibitor of metalloproteinase (TIMP) 1 in a wound healing environment (Figure 3A). The same effects were evident in keratinocytes, while the expression of TIMP3 expression was also slightly increased (Figure 3B). Further, a scratch assay measured the effects of α-13′-COOH on cells’ proliferative and migratory capacity after an injury. α-13′-COOH incubation of the fibroblasts significantly attenuates the scratch closure in the WHE (Figure 3C). However, in keratinocytes, the scratch closure was accelerated under the WHE condition after treating the cells with 2.5 µmol L^−1^ α-13′-COOH (Figure 3D).

Next, BNC-based wound dressings loaded with α-13′-COOH were studied. PEG400-formulated α-13′-COOH-loaded BNCs were produced. A continuous release over 24 h from the carrier material was observed (Appendix A). PEG400-loading of BNC increased the transparency of the fleeces, likely due to less interactions between nanocellulose fibers, leading to more considerable distances between fibers and enlarged pores. Higher transparency of the BNC dressing allowed the inspection of the wound during the treatment. Although the incorporation of PEG400 reduced fluid absorption and retention values compared to unloaded BNC, no adverse effect on wound healing was observed (Appendix A). Furthermore, loading with PEG400 led to constant moisture of the BNC over 24 h, thus preventing drying out of the dressing during the wound healing process. The mechanical stability, represented by the compression strength of the loaded BNC, was increased, which is essential during handling and fixation by second dressing materials (Figure 4A).

Following previous findings [25], BNC per se promoted wound healing (Figure 4B). The addition of α-13′-COOH to BNC did not affect the healing time of acute wounds (in comparison to BNC-PEG or α-13′-COOH alone) (Figure 4C,D). After 10 days, histological analysis of the wounds revealed a tendency of a thicker epidermis, fewer infiltrated neutrophils into wound tissue, and no signs of pathological collagen deposition (Figure 4E–G). The results suggest that in the splinted wound model, the healing process could be too fast to detect additive effects of the BNC when formulated with α-13′-COOH. Therefore, we aimed to test the α-13′-COOH-loaded BNC under diabetic conditions where wound healing is compromised due to persistent inflammation [40]. For these studies, streptozotocin, a toxin and GLUT2 transporter substrate, was injected into the peritoneal cavity. The preferential uptake of STZ damages β-pancreatic cells, resulting in hypoinsulinemia and hyperglycemia, and is therefore commonly used to induce diabetes [41]. In mice treated with STZ, blood glucose concentrations were analyzed over 14 days. All animals had constantly increased blood glucose concentrations and normal body weight development (Appendix A). After 14 days, splinted wounds were applied to the back of the animals and treated with vehicle (PEG400), BNC-PEG400, α-13′-COOH, or BNC-α-13′-COOH; wound healing and histological features of the wound were analyzed after 10 days. Again, α-13′-COOH significantly improved wound closure time over 10 days and led to complete wound closure in 5 of 6 cases (Figure 5B). In this model, the use of BNC-α-13′-COOH showed additive effects. Wound healing time was reduced further, and wound closure was observed in all cases (Figure 5C). The histological assessment of the wounds was nonetheless comparable in between all groups (Figure 5D–F).

## 4. Discussion

Non-steroidal anti-inflammatory drugs (NSAIDs) are recommended to reduce pain originated in ectopic wounds. However, studies indicate that NSAIDs also increase the risk of scar formation and might delay the wound healing process [42,43]. Their therapeutic effect is attributed to the inhibition of COX that otherwise converts arachidonic acid to proinflammatory mediators such as thromboxanes and prostaglandins [44]. In contrast, natural anti-inflammatory compounds, for example, α-tocopherols, can improve wound healing when applied topically [45,46,47]. The liver transforms α-tocopherol to bioactive, anti-inflammatory long-chain metabolites of α-tocopherol, such as α-13′-OH and α-13′-COOH, which limit inflammation by targeting 5-lipoxygenase in vitro and in vivo [21]. Moreover, α-13′-COOH exhibits anti-inflammatory properties by blocking the lipopolysaccharide-induced inflammatory response [48]. In our study, the α-tocopherol-derived long-chain metabolite α-13′-COOH and, to a lesser extent, the δ-tocotrienol derivative GA shortened wound closure time significantly when formulated and applied to ectopic skin wounds in PEG400 hydrogels using the splinted wound mouse model.

Besides the wound closure time, histological assessment of the wounds allows further conclusions about the quality of the newly formed tissue. α-13′-COOH and garcinoic acid accelerated the wound closure process and increased epidermal thickness significantly. In addition, the treatment further reduced collagen deposition, which indicates a reduced formation of scar tissue.

Notably, α-13′-COOH showed promising histological effects and speeded up wound closure. Treatment with α-13′-COOH reduced the amount of proinflammatory PGE_2_ and PGD_2_ in the wound tissue. Both prostaglandins are associated with impaired wound healing and reduced hair follicle neogenesis [49]. In contrast, α-13′-COOH treatment increased the concentrations of anti-inflammatory lipid mediators, namely resolvin D5 (RvD5), maresin 1 (MaR1), and protectin DX (PDX) in the wound. Several anti-inflammatory lipid mediators, such as resolvin D1, D2, and E1, were associated with improved wound healing [50,51]. Maresin 1 might support an anti-inflammatory milieu in the wound described after spinal cord injuries and bone fractures [52,53,54]. Ultimately, PDX might facilitate wound healing through the inhibition of collagen-depositing fibroblast proliferation [55]. This switch, characterized by decreased inflammatory and increased anti-inflammatory mediators, might be the consequence of α-13′-COOH that causes allosteric targeting of 5-lipoxygenase in the skin [56]. Further experiments on fibroblasts and keratinocytes in vitro indicate that the anti-fibrotic effects are not solely due to the altered local inflammatory milieu but also may be a consequence of direct anti-fibrotic and anti-proliferative properties.

Those data indicate that particularly the α-tocopherol metabolites, α-13′-COOH, exhibit positive effects on wound healing, shedding light on the complex anti-inflammatory properties of α-tocopherol.

BNC is a biocompatible, natural material with promising features as a wound cover and drug carrier. Providing a topical hydrophilic barrier and beneficially influencing the local wound milieu, BNC accelerates the wound healing process, and it reduces scar tissue formation and the risk of complications, such as infections [57]. Moreover, BNC serves as a fluid barrier that prevents local dehydration of the wound while interacting with the wound liquid and sequesters proinflammatory and pro-fibrotic cytokines, preventing increased collagen deposition, i.e., inhibiting scar tissue formation. Modified BNCs can create active wound dressings that allow the controlled release of active substances such as antibiotics that may provide additional beneficial effects in the clinical treatment of infected wounds [27]. Thus, combining the beneficial effects of α-13′-COOH with a controlled release from BNCs may be a promising therapeutic approach for treating complicated wounds. BNC loaded with PEG400 vehicle solution already significantly shortened wound closure time in the standard splinted wound model but resulted in no additional effects of α-13′-COOH-loaded BNC. However, histological tendencies, such as thicker epidermal layer and reduced infiltration of neutrophils, were observed for wounds covered with BNC loaded with α-13′-COOH. The skin has a high regenerative capacity, which metabolic comorbidities such as diabetes significantly limit. Those comorbidities delay the wound healing process and result in additional chronic wound infections that may become life-threatening. Thus, creating an antimicrobial barrier together with an inflammatory environment improving the healing process is desired. In our mouse model, we mimicked a diabetic condition and combined it to the splinted wound mouse model employing the well-established protocol of intraperitoneal streptozotocin injection, leading explicitly to pancreatic β-cell depletion and a diabetic situation delayed wound healing [41]. Comparing wound closure time between treatments with PEG400, α-13′-COOH, BNC, and BNC loaded with α-13′-COOH (all three of them loaded with PEG400 vehicle solution) in this model depicts the beneficial properties of α-13′-COOH. Regardless of the formulation, α-13′-COOH leads to a significant shortening of wound closure time. PEG400-loaded BNC had no significant effect compared to PEG400 treatment in this model. However, the combination of both, BNC loaded with α-13′-COOH in PEG400, leads to shortened wound healing times and complete wound closure, demonstrating an additive effect of the combinatory treatment.

## 5. Conclusions

α-13′-COOH is a naturally occurring bioactive vitamin E derivative, and this study depicts the potential for α-13′-COOH-loaded biocompatible nano cellulose to improve wound healing in complicated wounds. α-13′-COOH applied on wounds causes a shift in the inflammatory milieu of the wound characterized by an increased amount of anti-inflammation resolvins and impaired levels of proinflammatory lipid mediators. Furthermore, this shift in the inflammatory milieu is accompanied by a decreased infiltration of immune cells and a reduction of extracellular matrix deposition. Those histopathological effects correlate with an increased speed and degree of wound healing in the splinted-wound mouse model. Furthermore, the controlled release of α-13′-COOH from biocompatible nanocellulose wound covers maintained the positive effects α-13′-COOH in a controlled manner, which showed known positive effects on wound healing.

## Figures and Tables

**Figure 1 nanomaterials-11-01939-f001:**
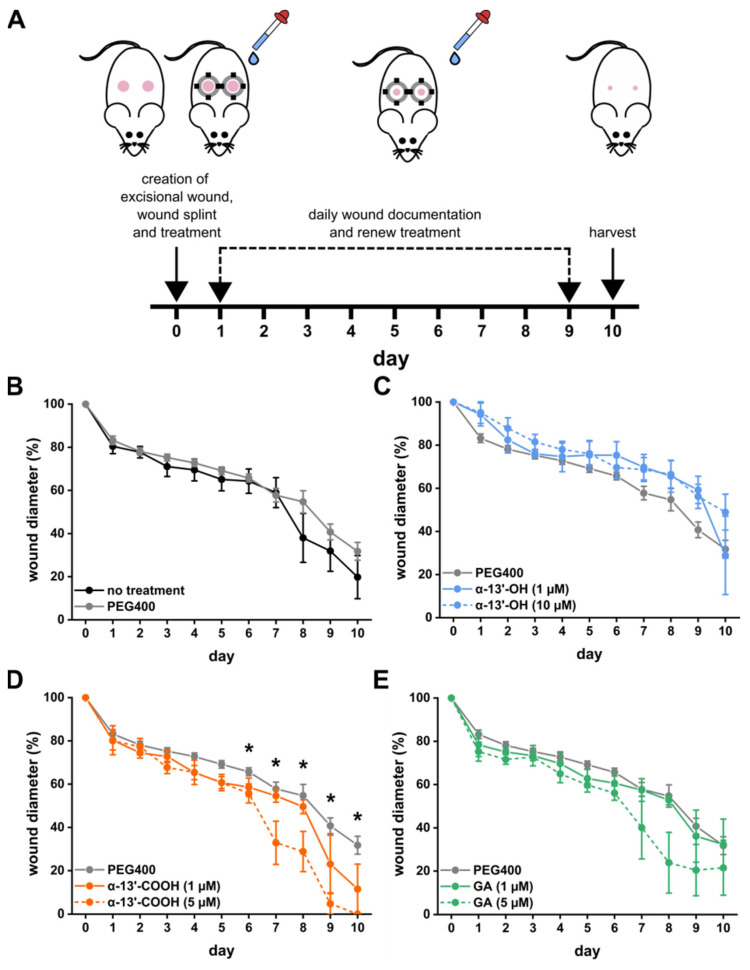
Effects of tocopherol and tocotrienol derivatives on wound healing. (**A**) Wound healing was assessed in the splinted wound model over 10 days. The wound diameters were measured daily to follow up the temporal wound closure when treating the wounds with (**B**) vehicle solution (PEG400), (**C**) α-13′-hydroxychromanol (α-13′-OH), (**D**) α-13′-carboxychromanol (α-13′-COOH), or (**E**) garcinoic acid (GA). All graphs depict the mean ± S.E.; n = 40 (PEG400), n = 8 (no treatment, α-13′-COOH (5 µmol L^−1^)) and n = 4 (other treatments); * unpaired t-test compared to wound diameter of PEG400 control on the same day, *p* < 0.05.

**Figure 2 nanomaterials-11-01939-f002:**
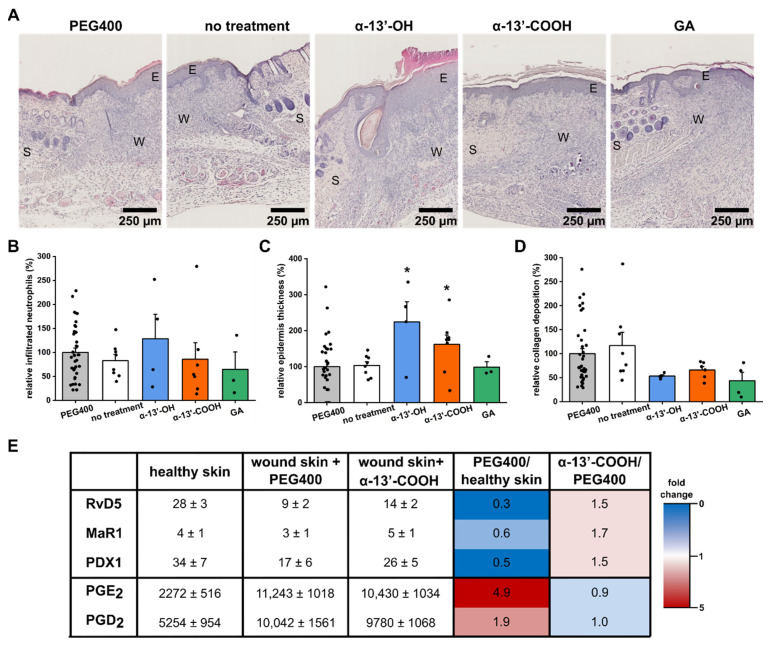
Histological analysis of wounds treated with tocopherol derivatives. (**A**) Histological assessment of wounds by H&E staining. W: wound; S: skin; E: epidermis. Scale bar: 250 µm. (**B**) Relative epidermis thickness was measured and compared to vehicle-treated wounds. (**C**) Infiltrated neutrophils characterized wounds after 10 days. (**D**) Scar formation, represented by collagen deposition in the newly generated tissue, is detected and analyzed by second harmonic generation imaging. Bars present the mean, whiskers the positive S.E., and the points individual analyzed wounds. * Unpaired t-test compared to PEG400 control, *p* < 0.05. (**E**) Formed lipid mediators in healthy skin or wounds treated with α-13′-COOH (5 µmol L^−1^) or vehicle (PEG400) were extracted and analyzed by UPLC-MS/MS. Data are provided in pg (as mean ± S.E.) in columns 2 to 4 and fold changes in the heat map in columns 5 and 6. n = 6.

**Figure 3 nanomaterials-11-01939-f003:**
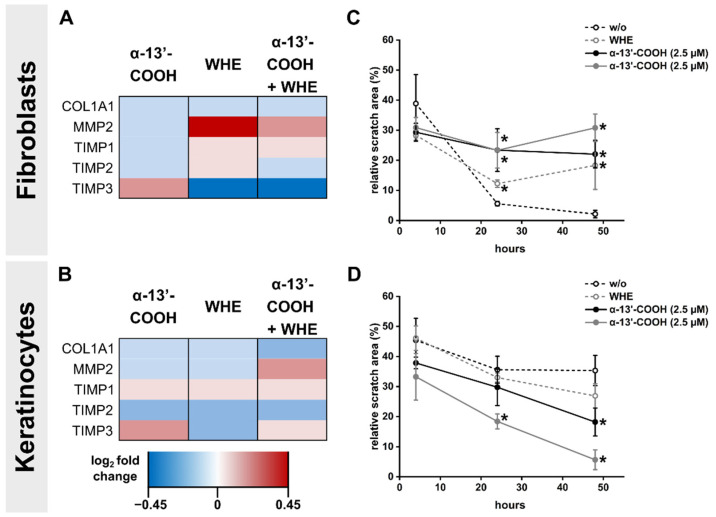
In vitro analysis of wound healing genes and proliferation in keratinocytes and fibroblasts. (**A**,**B**) Regulation of genes relevant for extracellular matrix measured in (**A**) fibroblasts and (**B**) keratinocytes treated with 2.5 µmol L^−1^ α-13′-COOH and WHE (wound healing environment: 5 ng mL^−1^ IL-1β and 10 ng mL^−1^ TNF-α) for 24 h. mRNA expression levels were normalized to RPL37A mRNA expression, which remained unchanged under all conditions. (**C**,**D**) Scratch assay with indicated concentrations of α-13′-COOH and WHE in (**C**) fibroblasts and (**D**) keratinocytes. Results are provided as a percentage of cell-free area to total recorded area. Data points depict mean, and the whiskers the positive and negative S.D. of n = 4 individual experiments; * unpaired t-test compared to control, *p* < 0.05.

**Figure 4 nanomaterials-11-01939-f004:**
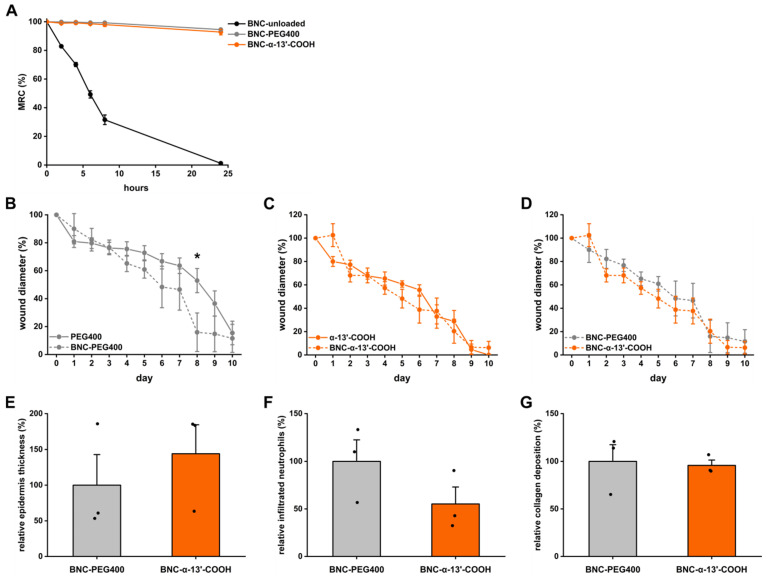
Controlled release of α-13′-COOH from bacterial nanocellulose (BNC). (**A**) Determination of the moisture retention capacity (MRC) of BNC fleeces by investigating mass loss at predetermined time points over 24 h. Fleeces were placed on a non-absorbent matrix without coverage. Unloaded BNC samples were compared to PEG400-loaded samples, and α-13′-COOH dissolved in PEG400-loaded BNC fleeces. Datapoints show the mean and S.D. of 2 separated experiments performed in triplicates each time. (**B**–**D**) Wound healing was assessed in the splinted wound model over 10 days. Wound diameters were measured daily. Data points represent the mean with the S.E. as whiskers of n = 4; * unpaired *t*-test compared to wound diameter treated with the solution on the same day, *p* < 0.05. (**E**–**G**) Characterization of the wounds by (**E**) epidermal thickness, (**F**) neutrophil infiltration, and (**G**) collagen deposition after 10 days. Data are represented as mean bar plots with S.E. and individual data points. * Unpaired t-test compared to BNC-PEG400 control, *p* < 0.05.

**Figure 5 nanomaterials-11-01939-f005:**
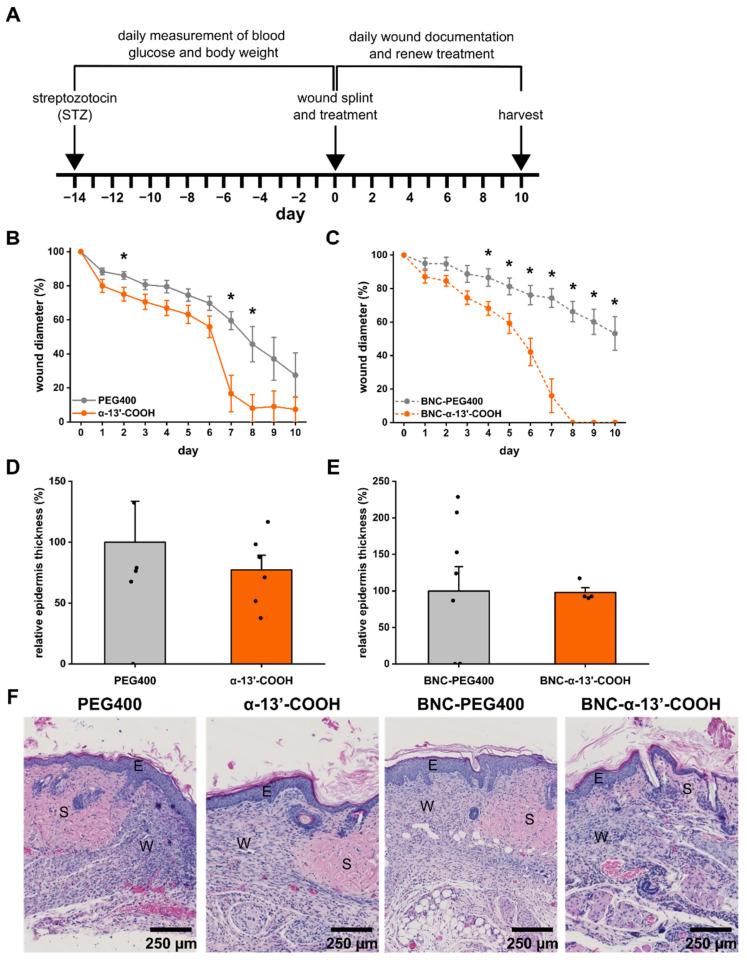
Effects of α-13′-COOH on splinted wound healing in the streptozotocin diabetes mouse model. (**A**) Experimental scheme. Wound healing was assessed in the splinted wound model over 10 days. Wound diameters were measured daily to follow up the wound closure time when treating the wounds with (**B**) α-13′-COOH and PEG400 vehicles or (**C**) BNC-α-13′-COOH and BNC-PEG400 vehicles. Data are represented as mean ± S.E. n = 5–9; * unpaired t-test compared to wound diameter of the vehicle group on the same day, *p* < 0.05. (**D**,**E**) Relative epidermis thickness was measured and compared to vehicle-treated wounds. Bars, whiskers, and points depict the mean plus S.E. and individual replicates. * Unpaired t-test compared to the vehicle group, *p* < 0.05. (**F**) Histological assessment of the wound was assessed by H&E staining. W: wound; S: skin; E: epidermis. Scale bar: 250 µm.

## Data Availability

The datasets presented in this study sorted by figures are available as a supplementary table to this manuscript. Further experimental information and raw data are available from the authors upon reasonable request.

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
