# Peer review of "Controlled Release of the α-Tocopherol-Derived Metabolite α-13′-Carboxychromanol from Bacterial Nanocellulose Wound Cover Improves Wound Healing"

_nanomaterials, 2021, doi:10.3390/nano11081939_

Round 1
Reviewer 1 Report
The authors present here study of α-tocopherol-derived metabolite from bacterial nanocellulose wound cover as a potential active wound dressing for the advanced therapy of skin injuries. It is interesting topic for investigation of effects of a natural product obtained from tocopherol-derived metabolite on wound healing activity under normal and diabetic conditions. However, this manuscript lacks some vital information. Also, the discussions for this version to support the big achievement of these fields are weak. Therefore, authors need the revision of the manuscript for publishing the work for publication in nanomaterials journal. Some questions and suggestions are followed;
1) The form of this manuscript does not match with the guideline of the nanomaterials journal. The authors should revise the form of this manuscript in the whole area with the guideline of the nanomaterials journal. The form of references described in References part does not match with the guideline of the nanomaterials journal. The authors should revise References’ form accurately.
2) We suggest that author should add the scale in figure 2a and 5f to increase understanding of this manuscript by journal readers.
3) We suggest that authors should change from 2.15 Statistics to 2.16 Statistics.
4) Conclusion section is ambiguous. We suggest that authors should add the detailed results as conducted by authors in conclusion section.
5) We judged that the title of main article and the title of Supplementary Material does not match. We suggest that authors should match the title.
6) We suggest that authors should correct the typographic error such as italic issue in the whole manuscript.
Reviewer 2 Report
A very interesting manuscript, addressing a very updated theme. Is will written, well presented and the results well discussed.
This reviewer only have some minor issues that think should be considered:
Introduction, pag 5 line 2 – The reference [1] appears two times.
Introduction, pag 6 line 2 – Chemically is not very accurate to say “…metabolites that differ in oxygenations…”. I suggest to rephrase it, with some like this “…metabolites that differ in the oxygen content of functional chemical groups…”.
Introduction, page 6, last sentence of the first paragraph- Please rephrase it to avoid the appearance of “…these derivatives..” twice.
Section 2.1, pg7, last line. Please rephrase the entire sentence, as it is written it seems that the described conditions were used only during the surgical procedure of the mice, which is impossible (12h light/dark cycles).
